# A Transition Period Ritual of the Karay Turks: Death

Emine Atmaca [1], Reshide Gözdaş [2], Ekin Kaynak Iltar [3,*], Rabia Akçoru [4] and Süleyman Ertan Tağman [5]

1    Faculty of Letters, Turkish Language and Literature, Akdeniz University, Antalya 07058, Türkiye; eatmaca@akdeniz.edu.tr
2    Faculty of Arts & Science, Turkish Language and Literature, Bolu Abant Izzet Baysal University, Bolu 14030, Türkiye; reshidegozdas@ibu.edu.tr
3    Faculty of Letters, Philosophy, Akdeniz University, Antalya 07058, Türkiye
4    Mediterranean Civilizations Research Institute, Mediterranean Medieval Studies, Akdeniz University, Antalya 07058, Türkiye; rabiakcoru@gmail.com
5    Faculty of Arts and Sciences, Philosophy, Burdur Mehmet Akif Ersoy University, Burdur 15030, Türkiye; setagman@mehmetakif.edu.tr
*    Correspondence: ekinkaynak@akdeniz.edu.tr

**Abstract:** Karaism is a Jewish sect that emerged in the Middle Ages and became the name of a Turkish tribe in time. Its name is derived from "kara- (K-R-A)", meaning "the ones who can read the sacred scripture" in Aramaic–Hebrew. The Karaites are members of the Jewish Karai sect, which only accepts the *Torah*. This feature naturally causes many differences. One of the main differences observed is the rituals for an individual in the death transition period, an important phase of human life. In this study, the death-themed core beliefs of the Karaites, which are brought from the roots of the Turkish genealogical tree, and the rituals that are combined with Judaism are analyzed. The differences stemming from geography and contacts with diverse cultures (such as Russian, Lithuanian, Polish, Belarussian, etc.) and the similarities in the rituals at the time of death and afterward stand out, especially funerals, which comprise the mourning traditions performed during and after the funerals belonging to the Karaites living in Crimea and Lithuania. Texts and words compiled from the Karay Turks living in Trakai, Lithuania, and the data acquired via observations are used to determine this information. In particular, the studies of Yuriy Aleksandroviç Polkanov, the head of the Crimean Karaites Association, are used for the data related to Crimean Karaites.

**Keywords:** Khazar state; Karay Turks; Karaism; Crimea; transition period; death rituals

## 1. Introduction

The Karai sect and the Karay Turks are topics that a few researchers could study further. Due to their inclination to privacy and their isolated lifestyles, there is not adequate information on the origin of the Karay Turks, their reason for choosing Karaism, or simply the way they carry out their rituals. In this study, we aim to find answers to some of these questions according to the information we have gathered and analyzed. Firstly, the Karay Turks, who owe their sect and national equality to the general ethnogenesis of religion and language, are discussed using historical data. Then, both the death-themed essential beliefs of the Karaites in Crimea and Lithuania, which stemmed from the root of the Turkishness tree, and the rituals blended with Judaism are analyzed together. During the analysis, the differences that originated from contact with different geography and cultures drew attention, as well as the similarities between the death transition period practices of the Karaites. Field research was carried out when gathering this information by using observation and interview techniques with three people residing in Trakai, Lithuania, whom we consider as a source and assigned identification numbers such as 1, 2, and 3 in the main text. Whether there are similarities between the customs, beliefs, traditions, and contemporary practices and whether these rituals still endure based on the information given by

the three people who were interviewed during the field research were established. Occurrences/situations were examined in their natural flow, and notes were taken constantly. The observation was carried out objectively without subjective evaluations. Lastly, the works of Yuriy Aleksandroviç Polkanov, Crimea Scientific Council President, were considered particularly fundamental for the data obtained from the Crimean Karaites.

**2. The Karaite Sect**

Religion is a sociocultural system that has persisted for thousands of years in terms of human history to this day. Judaism is among the oldest religions in human history and the first example of a monotheistic religion. It is an ethnic religion that consists of the beliefs, culture, moral structure, and laws of the nation of Jews. It is divided into sects in time with the impact of constantly changing discourses: Hasidics, Pharisees, Sadducees, Essenes, Jewish Christians, Zealots, Yudghanites, Rabbinic Jews, etc. (Küçük et al. 2012, pp. 331–39). One of them is Karaism or the Karaite sect, which emerged from Judaism due to influences from religious, political, and social causes.[1]

The etymology of the word "Karâîm" ("Karayim" in Turkish) is related to the Ancient Hebrew root "kar" קרא, meaning "to read". From this root, the participle *karay* קראי, meaning "the one who reads", is derived. This participle is conjugated in the plural as קראים = Karâîm, meaning "the ones who read". While the word "Karâîm" is widely used to represent a religious group of Karays all over the world, the religious belief of this group is called "Karaism". The name "Karay" appears among people of Turkish descent and in literary language as *Karaylar, Karay, Keray*, and *Karait* (Мусаев 2003, p. 7; Kocaoğlu 2017, pp. 7–8). Additionally, from this root, the word *mikra* is derived (the Sacred Scripture or the Old Testament). This is the only sacred book of the Karaites (Kefeli 2002, p. 4). Those who are disciples of the Karai sect accept and read the Tanakh (~the Old Testament) as the source of religious commands (2). For the Karaites, Tanakh is written clearly, and it is free from mistakes (Дубинский 2005, p. 50). Tanakh is an acronym comprising three sections of the sacred book of Judaism: *Torah, Neviim*, and *Ketuvim*'s capitals (Kefeli 2002, p. 8).[2] Tanakh consists of 24 chapters that are divided into three books: "Law", q.e., *Torah*; "Prophets", q.e., *Neviim*; "Scripture", q.e., *Ketuvim*. The first of these is *Torah* or *Tevrat*,[3] and *Torah* means "code, shari'a, law, display, education, doctrine". It is divided into two parts: *written* and *oral* (Talmud). According to the faith, God gave written and oral *Torah* together as the oral one is an explanation of the written one to Moses at Har Sinai (~Mt. Sinai) or Mt. Horeb. Moses reported this orally to Yeshua, who would become the head of the tribe after him (Kaya 2015, pp. 81–82). The written *Torah* consists of five books, and it was given to Moses with its supplements by God at Mt. Sinai. Oral *Torah*, q.e., *Talmud*, is a written and supplementary book that consists of words attributed to Moses, which are accepted as his expressions, and *Torah*'s commentary (Kutluay 2001, p. 164).

The Karaites only believe In the written *Torah*, and the Ten Commandments in *Torah/Tevrat* are the foundations of their faith. One of the most important Karay Turks, Avraam Ben Shemuel Firkoviç, mentioned this matter in an interview: "The foundation of the Karay religion is the Old Testament (Дубинский 2005, p. 40), especially *Decalogue or Aserot ha-Diberot* (~Ten Commandments). Ten Commandments symbolize the part to be fulfilled by the Israelites, which is from the testament made through Moses between God Yahweh and the Israelites (Atasağun 2001, p. 136). Therefore, the firm commitment to the Ten Commandments, including honesty, modesty, and clemency, is considered a crucial moral responsibility for the disciples of the Karâîm sect. There are "Ten Dogmas of the Doctrine" developed at the end of the Middle Ages with the *Decalogue* (Дубинский 2005, p. 41; 2). The Karaites reject *Talmud* in which the productions of the oral tradition are to be found and accepted as the foundational text of the Rabbinical law. Since Talmud includes later additions which they consider to be against the written law" (Firkovičius 1994, p. 34; 2). As observed, the Karaites are strictly committed to the original transcript at the time of the Prophet, which is called *Ветхий Завет/Старый Завет* or "Old Testament". Thus, they comply with this commandment in the *Torah*: "You shall not add to that which I command

you and you shall not subtract from it, but keep the commandments of the Lord, your God" (Eliade 2019, pp. 205–6). This transcript is of divine origin and no interpretation, explanation, or any other modification was added to the transcript (Kokizov 2011, p. 875). Moreover, the Karaites changed the prayer that, resembling the Islamic confession of faith, Jews recite constantly in every ritual as "Shema Yisrael, Adonai eleheinu, Adonai ahat".[4] Instead of addressing Israel, they say "Shema Karai, Adonai eleheinu, Adonai ahat"[5] (Tanyu 1978, pp. 34–54).

The Karaite sect, which initially emerged as a Middle-Eastern-based religion, appeared in many regions of the world by particularly spreading to Palestine, Syria, Egypt, Byzantine, Iran, Armenia, and Caucasia (Danon 1925, p. 289; Suleymanov 2012, p. 22). The sect was established and systematized in c. 760 by Anan ben David (715–795) of Basra in the Abbasid Dynasty, even though its roots are traced far back in Jewish history. Rabbanites also date the sect's foundation to Anan ben David in the 8th century, and they view them as their opponents (Lasker 2011, p. 427). This sect, which is named Ananiya after its founder in the beginning (Kuzgun 1985, pp. 156–57), is named the Karaite sect after a short period (Doğruer 2007, p. 38).

Although the first Karaite community appeared at the time of Anan ben David, Talmudistic Jews, especially the ones in Iraq, objected strongly since the Karaites objected to the interpretation of Tanakh, similarly to the Talmudists. At the beginning of the 8th century, Karaite communities emerged in many places, especially in Jerusalem. One of them is Khazar country where most of the population consists of Turks (Şişman 1957, pp. 97–102). Unfortunately, sources on this subject are scarce, although research of great importance was carried out on the Khazars' conversion to Judaism. One of the most important sources is an anonymous epistle, which was introduced to the scientific world in 1912 and named after its publisher as *Schechter Epistle* or after its location where it was found as the *Cambridge Document* or *Kenize Epistle* (Karatay 2008, pp. 1–17). The Khazar state acknowledged Judaism between 750 and 790 according to this source (Suleymanov 2012, p. 50). Two papers on the Khazars called *Khazars* and *The Khazar Khanate* in the latest edition of *Bolshaya Sovetskaya Ansiklopedya* contain this conclusive statement: "Towards the end of the 8th century, the strata of the Khazars accepted Judaism (Zajaczkowski 1961, p. 480). It is known that the Karaite sect emerged in the Khalifate of Abū Jaʿfar ʿAbd Allāh ibn Muḥammad al-Manṣūr (754–775) engaged in missionary activity at that time and converted the people of other countries. The Karaite missionaries arrived at Caspian and Black Sea steppes through Byzantine, and here they made the sect spread among the Khazars, the Kumans, and other Turkic people (Zajaczkowski 1961, p. 479; Зайончковский 2005, p. 69).

## 3. The Karay Turks

The Karaites are the only people in the world who have lived in Crimea since Antiquity and accept the Karai sect as their national religion (Бабаджан et al. 2000, p. 19). Their immigration to Crimea took place long before the Kipchaks, Khazars, and Tatars appeared in this region. The Karaites have lived together with the Tatars in Crimea. Tatar Khans frequently gained the support of the Karaites while struggling with each other. Even the Karaites supported Tokhtamysh Khan over Mengli Giray, and they helped him to be triumphant in the battle against Giray. At that time, the Karay Turk community consisted of only 40 families.[6] The name "Kırk-Er" or "Kırk-Or"[7] (40 places) was given to a castle in the dedication of their victory, and a piece of marble with engraved victory symbols, such as stirrups, pitchforks, and shields, was hung on the main door of the castle. These symbols are also preserved to this day (Lebedeva 2003, p. 5). Bertier-Delagard provided information about Chufut Kale (~Kırk-Er/Kırk-Or) as follows: "/.../ It existed long before the Tatars. Whatever the name was given, this city was so important in the country that when the Tatars attempted to siege it, the siege lasted longer than it could be, which meant the opponents' defense was persistent". After capturing the castle, the Tatars placed their garrison inside during the struggles for the independence of the Crimean Khans against the Golden Horde and the civil wars of the Giray Dynasty (Lebedeva 2003, p. 6). At the end of the 13th

century and the beginning of the 14th century, these civil wars in Crimea forced some of the Karaites to immigrate from Crimea to the Principality of Galicia–Volhynia and Lithuania. Another reason for this immigration decision was that the Grand dukes of Lithuania were called by Gediminas (1275–1341), Kestutis (1297–1382), and Algirdas (1296–1377) to the Grand Duchy of Lithuania for military service and to battle against the Teutonic Knights. These dukes conferred freedom upon the Karay Turks and permitted them to settle on the shores of the Vokė River near Vilnius. The Karay Turks began to settle in Lithuania in the 14th century, especially during the reign of Prince Vytautas (~Witold) (1392–1430). Prince Vytautas placed 383 Karay families from Crimea (near Old Crimea, Eski Kırım) in Trakai in 1398 (Мусаев 2003, p. 7). Thus, they owned lands that they forever benefitted from, on one condition that they should never sell their lands. In exchange for land, they owed military service to the dukes (Grişin 2000, p. 7).

The Karaites, who are ethnically considered Turkish people (Şayhan 2012, p. 95), describe themselves as *Karay* or *Karays* and trace their origin to Khazar Turks (Dunlop 2008, p. 12; Grişin 2000, p. 19; Şişman 1957, pp. 97–102). The Karay Turkish tribe descended not only from Khazars but also Bulgarians, Uz-Pechenegs, Kipchak Cumans (~Comans), and the old Turkish tribes: *Uzon, Çuyün*, and *Nayman*'s interactions were inferred from this Turkish tribe's texts according to native and foreigner Turcologist research, even though it is recorded that the Karaites are a Turkish tribe consisting of the Khazars. An Abbasid embassy delegation was sent to the Itil/Volga Bulgar Khanate, and as the delegation scribe, Ibn Fadlan's testimony in his *Risāla* proves this information. The testimony is as follows: "the Khazar Khanate consists of 25 people each of whom sent a bride to the Khanate as its subject". This community of Turkish descent accepted Karaism, which bore the same name as the sect they belonged to. Ibn Fadlan provides the following information: "all of the Khazars and their rulers are Jews"; however, there is no evidence supporting that they visited the Khazar State in his *Risāla* (Fadlan 2017, p. 47). In contrast, it is claimed in historical data that Khazaria was a cosmopolitan country where many people from diverse nations and races came together, and in the capital city, of Khazaria, Itil (~Sarkel), there are four different religious systems, such as Christians, Muslims, followers of the *Torah*, and ones who did not follow a scripture (q.e., Kamlik religion/Shamanists) (Zajaczkowski 1961, p. 479; Golden 2006, p. 285). Furthermore, religious diversity in the state's legal system stands out. As narrated by the Arabic geographer Mesudî, there were seven qadis in the capital of the Khazaria, Itil: Two of them were Christians, two were Muslims, two believed in the *Torah*, and one did not follow scriptures (q.e., Kamlik religion/Shamanists) (Zajaczkowski 1961, p. 479; Golden 2006, p. 285). Historical data show that the patrimonial class and intellectual class adopted Judaism, whereas the people were still following different belief systems (Türkdoğan 2011, p. 102). This example of *convivencia* could be explained as the indulgence of the Khazar Khan to his people (Zajaczkowski 1961, p. 479).

The Turkish traveler Evliya Celebi claims that the Crimean Karay Turks living in Crimea were different from Jews with respect to their faith (Бабинов 2004, pp. 11–12): "All of the Jews belonging to the Karai sect are Chufuts. Other Jews do not like these. They do not seek what is kosher (halal) or forbidden (turfa) in their foods. No matter whose food it might be, whether it is fatty or not, they eat it. These are the Qizilbashs of Jews and at the apocalypse, they do not ride the Qizilbash, but the other Israeli Chufuts do. They say 'Râfizî rûz-ı kıyâmet har buved zîr-i yehûd' as a verse. Those are Israeli Jews. They read Torah and Psalm, but they never know the Chufut language. They speak the Tatar language, and they wear Tatar kalpaks from purple çuka,[8] not hats /…/" (Kahraman 2013, vol. VII, p. 266).[9]

There is no doubt that Seraya Shapshal, the fourth hazan of the Crimean Karaites, had an immense effect on identifying the ethnic identities of the Karay Turks. Shapshal attempted to prove and convey his thesis that the Karaites had Turko-Khazar origins in his pamphlet titled *Karaimı i Çufut (Çuft) Kale v Krımu* (*The Karaites in Crimea and Chufut Castle*), which was published when he was a student in the faculty of oriental studies of

St. Petersburg University in 1896. The pamphlet has two main sections. He discussed the emergence of the Karaites in Crimea; their origin; their first settlement; the Karaite sect; the difference between the Karaites sect and other Jewish sects; spiritual leaders; the emergence of Russian Karaites; and so on in the first eleven pages of the first section. The settlement of Chufut Castle; places to see around the castle; the walls and door of the castle; temples; shrines; the living conditions of the Karaites near Chufut Castle before the Russian rule in Crimea; the fate of Chufut Castle since 1783; prominent people who visited the castle; the old cemetery of the Karaites; the speech of the Rabbi presented to the Tzar Alexandre III; and so on were discussed in the second section (Şapşal 1895, pp. 1–28). This brief but valuable work of Shapshal comprised the beginning of the research concerning the history of the Karaites, their ethnicity, their culture, Chufut Castle, and the Karaite sect. Later research has revealed that the Karaites adopted a religion of their own within Judaism, named Karaism, which accepts the *Torah* as the only scripture and rejects Talmud; by this aspect, the Karaites have different beliefs from Rabbinic Jews following Talmud. The Karaites have their place both in Judaism by believing in the *Torah*, Hebrew scripture, and Moses and in Turkish religious and cultural history due to their origins as Turks. As in other Turkish communities, the Karaites maintained traditional Turkish religious beliefs and lifestyles (Arık 2005, p. 49).

Daniel J. Lasker, a professor at Ben Gurion University known for his studies in Karaite Judaism, has different claims regarding the formation of the Karaite ethnicity: "A good example of the new Russified, secular, nationalist Karaites was Seraya Shapshal (1873–1961), the man most responsible for Karaite 'de-Judaization'… There is no doubt that Shapshal drew some of his inspiration from the Turkish nationalism of Kemal Ataturk. Part of Shapshal's reforms were to replace Hebrew with the Turkic Karaim language wherever he could, e.g., in the synagogue liturgy and on tombstones in the Karaite cemeteries. The language was now written in Latin letters, no longer in the traditional Hebrew characters. Loan words, especially from Hebrew, were replaced by newly coined Turkic words. Hebrew personal names were abandoned, and a new calendar, with the names of the months and the holidays replaced by 'Turkic' names, invented by Shapshal, was devised. Shapsal rewrote Karaite history as well. Suddenly, Jesus and Muhammed were considered Karaite prophets. Karaites were said to worship sacred oaks in the Chufut-Kale cemetery, thus turning Karaism into a syncretic religion that included elements of Judaism, Christianity, Islam, and paganism. Shapshal also created a coat of arms, with pagan symbols, to replace the iconography of the Star of David and Ten Commandments in Karaite synagogues. The word 'synagogue' itself was banned and replaced by the less-Jewish-sounding term 'kenesa' or 'kenasa'. Most importantly, the Karaites were declared to be descendants of Turkic tribes, with no historical affinity whatsoever to the Jewish people" (Lasker 2020, p. 6).

The words of Szymon Juchniewicz from the Lithuanian Karay Turks regarding religious authenticity are worth mentioning. During our interview, although we did not pose any questions regarding this matter, he expressed these words repeatedly: "We are definitely not Jews even though our prophet is Moses, we are Turks, our traditions are the same as yours, we speak Turkish like you". We gathered text and word compilations from Juchniewicz and Hazan Lavrinovicius, a great deal of which are in the Turkic Karaim language (Troki dialect) rather than Russian. Hazan Lavrinovicius stated that both Jesus and Muhammed were subjected to the laws of Moses; they believed that the genuine book of religious law was the *Torah*, and neither the Quran nor Bible abolished the *Torah*. Moreover, he mentioned that they considered the Psalms of David valuable, and they prayed using the Psalms during daily rituals.

The Karay Turks owe their national equality to the ethnogenesis of religion and language. This situation united these Turkic people, ensured the preservation of national unity, and at the same time transformed the name of the religion into an ethnonym (Zajaczkowski 1961, pp. 480–81). Language has a common national value for this ethnic group. Therefore, the mother tongue equals national identity and religion for the Karay Turks. Within this respect, Crimean Karay Turks described their ethnic identity in the

Ukrainian Congress in 2003: "Crimean Karaites are the natives of Crimea who are unified through common bloodlines, language, and tradition. Crimean Karaites are fully aware of their blood relations with Turkic peoples, their ethnical privileges, their cultural authenticity, and their religious independence, they also feel unique feelings towards Crimea, for it is their historical homeland" (Suleymanov 2012, p. 44). In the same manner, during our interview with Szymon Juchniewicz, a Lithuanian Karay Turk, he mentioned this sentence repeatedly: "We are not definitely Jewish, we are Turks, our traditions are the same as yours".

As a historian of religions, Ahmet Hikmet Eroglu provided information that the Karay Turks living in Istanbul are different from other Jews in his work titled *Jews Among the Ottoman Turks (Until the end of 19th century)*: "Jews have another cult. It is called Karaim. The members of this cult differ from ordinary Jews. They do not live together with other Jews. They have their own unique houses.[10] They do not eat or drink in the same place, they feed their animals separately, and they make their wine. Their diets are specific to themselves. While other Jews eat meat, they eat fish. The members of this cult do not intermarry with others. They are committed to the five books of Moses and the Ten Commandments" (Eroğlu 1997, p. 86).

### 4. The Transition Rituals

In some cultures, the three stages of birth, wedding/marriage, and death are considered as the transition periods of human life. The customs, traditions, rituals of these transition periods, and their applications constitute the fundamental stages of Turkish tribes' traditional culture. The Karay Turks also consider the transition periods important, similarly to other Turkish tribes, and they apply the practices by themselves according to the remnants of the old Turkish religion and Karâî belief system.

The third turning point of the transition period naturally is "death". There are various beliefs, traditions, customs, and rituals about death among the Karaites as well as among other Turkic peoples. The foundation of these practices was formed by Tengrism or the traditional Turkish religion (Мусаев 2003, p. 9). While applying the practices, the deceased is feared because of the various negativities caused by death, and souls are deeply respected. This procedure comprises preparation for the deceased to be sent to the afterlife, and the idea that the deceased is still described as living and the need for protection from them lies behind this procedure. Moreover, concerns about hygiene and especially religious traditions are important factors (Örnek 1995, p. 214). For the Karay Turks, religious customs and traditions are more important when helping the deceased transition to the other world; in the process of transitioning, the soul of the deceased peacefully transitions into their new life (2). It is essential to provide protective power from the ancestors' cult, which comprises respect and fear when performing these rituals.

### 5. The Death Rituals of the Karay Turks

The Karaites' death customs and rituals, which are carried out to prepare the deceased for the other world, are very different from the Jews' customs. As mentioned before, many symbolic rituals of the transition periods of the Karaites consist of traces belonging to the traditional Turkish religion (Altınkaynak 2006, p. 12). Moreover, it could be claimed that this belief survives in new forms, even though they have adopted another religion or sect (Suleymanov 2012, p. 97).

There is no music in the worship and funeral rituals of the Karaites (3). They chant hymns and an elegy for the deceased, and they sing songs. To bury the deceased and carry out lustration, a kind of shroud and prayer by Hazan is necessary (1; 2). By carrying out these three processes, there are many traditional beliefs and religious rituals taken from the Karâî sect, so much so that most of the time, these beliefs stand out more (2).

Anyone or a messenger in the Karay community informs everyone about the deceased, similarly to weddings. The messenger offers their condolences by saying "bashin sav bolsun"[11] at every house upon arriving. The messenger gives information about the

identity of the deceased after having received the expression "dostlar sav bolsun"[12] in return. Since it is forbidden for the relatives to touch the deceased in Karaism, burial services are carried out by special officials who are called gabars (Suleymanov 2012, p. 127). Burial services are often carried out by two men or two women. Some representatives of certain families were the only individuals that could be gabars long ago; however, at the beginning of the 20th century, assigning Karaites that have poor financial situations as gabars have become a tradition (Чижова 2003, p. 80). Gabars in Crimea typically live in a house near the kenesa/prayer house (Çulha 2012, p. 396); this does not apply to the Karaites in Lithuania (1).

It is forbidden for the Karaites to shake hands, hug, and kiss at the mourning house and cemetery. The deceased needs to be buried as soon as possible according to the Karaites (Чижова 2003, p. 80). Burying the deceased on Saturdays (~sabbath "resting day")[13], Rosh Hashanah (~new year), Yom Kippur (~boşatlık "day for redemption"), Pesach (Passover), Shavuot, Sukkot (~Sukott), Simchat Torah, etc., is also forbidden. Therefore, the deceased is kept in a place called the ghusl room in kenesa (Çulha 2012, p. 396). After the person is accepted as deceased, a sheet is laid on them. A coffin is brought, and the corpse is washed on a special board, which is then brought to the house of the deceased by the cemetery guard (1). This board is round, with low iron-clad rims (6–8 cm), and it has holes for water discharge (Чижова 2003, p. 80). The cemetery guard is the one who brings the board (1). Having been washed, the deceased is clothed in white and shrouded with a white shroud. If the deceased is male, underpants, a long linen shirt, socks, a bathrobe, slippers, and a hat are used; if the deceased is female, a shirt, pants, socks, a dress, slippers, and a hijab are required (Saraç et al. 2007, p. 163; Suleymanov 2012, p. 128).

A velvet cloth is placed in the coffin, and over the velvet cloth, a clean white sheet is laid; on top of them, the deceased is placed with stretched arms, and a little pillow sits under their head. Soil from the family cemetery or Balta Tiymez cemetery in Crimea is placed over the eyes of the deceased. Again, a velvet cloth or a regular sheet is placed over the deceased. Their body and face are covered separately with a handkerchief. It is important not to place any jewelry anywhere near the deceased (1). After closing the coffin, a black cloth is placed over it (Polkanov 1994b, p. 56). Before taking the coffin out of the house, relatives and friends place pieces of fabric (velvet, silk, etc.) over it, of which their quantity and quality depend on their financial situation. These pieces of fabric are given to the closest relatives of the deceased, and they are handed out to Hazan and the poor after 7 days (Чижова 2003, p. 81). If the deceased is female, a marriage contract called şettar[14] wrapped with white linen is placed on the coffin. It is a custom among the Karaites to carry the deceased in a closed coffin because "the open coffins of the deceased at the funerals are profane"; this is accepted as a rule by a special decision of the Karay clergy congress (Бабинов 2004, p. 16; Suleymanov 2012, p. 128).

Concerning prayers, two candles are placed near the head of the coffin during the night. Then, Hazan recites kıyna, which are the mourning songs for this special occasion, and zeher is recited, which is a kind of prayer for the dead. The prayer is usually recited in Karay Turkish; however, in the 20th century, it was common for it to be recited in Russian (Polkanov 1994a, pp. 29–30; Чижова 2003, p. 81). As for Lithuanian Karaites, two or more people recite prayers for the deceased until the morning after Hazan completes the recitation. It is believed that sins are forgiven by God without the agency of a religious leader. Consequently, none of the Karay Turks speak loudly of their sins. Therefore, when praying, they ask for forgiveness from God both for themselves and for the deceased (2). They recite prayers from the Psalms of the prophet David (Kobeckaitė 2006, p. 11) and a book titled *Karaj Dińliliarniń Jalbarmach Jergialiari*, which translates into "Supplication Texts of the Members of the Karay religion" (2). The main components of the prayers of the Karaites are separate parts of the Psalms and hymns written by the clergy. In general, prayer consists of four parts, which are praising, expressing gratitude, repentance, and supplication. The rituals of the Karaites, including the order of the prayers, were formed at the end of the 13th century (Дубинский 2005, p. 42).

The coffin's lid is always kept closed during prayers. If the lid is open, the visitors leave money on the stomach of the deceased; if it is closed, they leave it on the coffin's lid. Hazan prays for the funeral in a similar fashion to the salah in Islam before taking the deceased out. Among the Crimean Karaites, this prayer is recited in Russian today, and it used to be recited in Karay Turkish (Suleymanov 2012, p. 129). On the other hand, in Lithuanian Karaites, this prayer can either be recited in Karay Turkish or Russian (2). After Hazan prays for everyone, different prayers for the identity of the deceased are said. Even though the variety of the prayers depends on whether the deceased is old, a religiously respected senior, a single young man, a married man, a single young woman, or a married woman (Koçak 2015, p. 427), Hazan usually prays as follows: "Every living being's life and breath is under God's rule. All the doctrines of the earth are under God's rule and the power of the mountains comes from him. The sea he created is his. The land his power created is his. He never discriminates between the rich and the poor he created by his power. Everybody eventually returns to the ground. All of them lie over the ground and are covered by worms. God is the one who gives power to everyone, both the one who makes mistakes and the one who deceives. Mind and courage are given to all. A wise and powerful person never says anything unpleasant. He is the one who tears down the mountains, but they never know it is God's rage. He is the one who makes the land tremble, but humans come up with other causes for it" (Firkovičius 1999, p. 107; Koçak 2015, p. 427).

Then, Hazan continues their prayers as follows: "God is true and rules with justice. Praised be God whose every order is true and fair. If God stops and thinks, what will I say as an answer? Therefore, I fear your presence. I think and fear him. Lord, sitting on the throne of justice, I am purged from my sins, and who says, 'I purified my heart' is gracious to all sins" (Firkovičius 1999, pp. 107–8; Koçak 2015, p. 427).

Before the coffin is taken out of the house, Hazan prays as follows: "The mourning is heard with a wail/ The person fades into the distance from us/ The soul scatters and shakes/ The eye may shed sorrowful tears/ They left this world/ They went to their eternal place/ The sufferings they had/ Had shortened their life./ May you keep crying and groaning/ Lamenting too/ May you comfort the ones mourning/ They were orphaned./ Protect your flock my God/ Take away their worries./ Bring together what has been scattered/ Give comfort./ Spare your servants/ Guarding your unity./ Send them your forgiveness/ May they find power in your presence/ Send them your forgiveness/ May they find power in your presence" (Firkovičius 1999, pp. 111–12; Koçak 2015, pp. 427–28).

It used to be customary to keep the coffin in kenesa, a Karaite synagogue, for a while. However, it has become necessary to carry the coffin to the cemetery without delay after the prayer is carried out (1). During the funeral procession, Hazan follows the coffin closely, and the relatives follow the funeral procession at a certain distance. It is a tradition for the funeral procession to cease for a short moment when passing by shrines in the kenesa (Чижова 2003, p. 81). If the house of the deceased is close to the cemetery, the coffin is carried over the shoulders; if the house of the deceased is far, then it is carried over the shoulders to a car, which then takes it to the cemetery. Hazan follows the coffin, and the relatives of the deceased follow Hazan (Polkanov 1994a, p. 41; 2). They pay attention to taking the coffin out of the house with its head first since it is believed that a human being leaves the world the same way they came into the world (1). If the funeral passes by a house where a Karay lives, then the owners of that house must keep all windows and doors closed (1). The windows and doors are kept closed to protect the household from various negativities caused by the deceased and deceased souls (2). In the cemetery, Hazan recites a kind of salah and prays. This mourning comprises a prayer that narrates the deceased's life. This prayer is called syjyt jyry, meaning "the song of mourning" (Kobeckaitė 2006, p. 11; Firkovičius 1999, p. 129).

The grave is dug in the north/south orientation. The coffin is placed in the grave, with the feet of the deceased facing the south and their face oriented toward Jerusalem (2). The deceased must be buried in this fashion so that they face Jerusalem (1). The kiblah

orientation for the Karaites is oriented toward the temple of Solomon in Mount Zion, q.e., Beit Hamikdash (~Western Wall) (1).

Hazan prays while facing the grave: "Here even the servant is liberated from their servant. Generations come and go, but that place stays for eternity. What benefit is there for the person who suffers under the sun throughout their life? How will the end of mankind and animals be? They are not different from one another. All have one life, and their deaths are alike. If the whole world gathers in the same place, death will be the same for the faithful, the blasphemous, the good and pure, the villain, the innocent or the sinner, the one who took the vow, and the one who fears the vow. May you never utter evil words, and may your hearts never want to speak disgracefully in the presence of God, because God is in the heavens, and you are everywhere. Therefore, speak less. Try to do the best that is in your power to do, because your plan or mind does not matter in the grave which is your last resort. I looked and have seen that it does not matter what those who live under the sun do, whether it is that the one with the feet walks, the one with the courage builds an army, the one with the mind eats bread, the wise one teaches and the one who knows how is loveable. Everyone has a certain time no matter what is done, and everyone goes towards a common end. Humans do not know when death will come. Mankind is trapped as the fishmongers catch fish or the birds are trapped, and suddenly it finds itself in that bad moment. God turns all he created, and all their secret sins to his judgment to give reward or punishment for their good and bad deeds. These are the words to say besides those which you heard until now: 'Fear God and do his bidding because being human requires this'" (Firkovičius 1999, pp. 110–11; Koçak 2015, pp. 425–26).

Hazan and those present at the funeral throw a handful of soil over the coffin. It is improper to pass the shovel from hand to hand when throwing soil; the shovel should be left on the ground. The blood relatives never touch the coffin and never throw soil over the deceased with the shovel (1). A stake is piled at the two sides of the grave (Polkanov and Polkanova 2005, p. 107). This type of piling indicates the gender of the deceased person in the grave. Spears were commonly placed in the ground near the grave in the past (1). If the grave has a spear or stake, the deceased is male. If the grave does not have a spear or stake but has a stone or a branch both at the head and feet area, this grave belongs to a woman (1). Then, a rock or a branch of a tree over the grave is placed at the feet area (2). After the burial procedure, the grave is cleaned, and Hazan prays for the last time as follows: "Mighty God of the whole world! You, who are merciful, benevolent, and kind, are loved. Have mercy on the beloved deceased, …, who left us today, son of …/ beloved woman ….'s husband (daughter) …/ young ……'s son …/young girl …'s daughter's …/ young boy ….'s son (daughter). Amen" (Firkovičius 1999, pp. 112–13; Koçak 2015, p. 426). "Our almighty God, remember them with their people's love and think of them with goodness. Strengthen their place in heaven. Comfort the hearts of those who mourn ardently. God of the whole world, give them comfort for their pain, turn their anxieties to joy and heal their wounds. God of the whole world, drive away the death, calamity, destroyers, various fears, and anxieties from us, our homes, and all our people in honor of your mighty and sacred name. Your mercy is abundant. Let's say 'Amen!' to you. There is no doubt that God will be praised until eternity (Rest in peace)" (Firkovičius 1999, pp. 112–13; Koçak 2015, p. 426).

The Lithuanian Karays are buried in cemeteries in Trakai, Vilnius, or Naujamiestis (Kobeckaitė 2006, p. 11). As for the Crimean Karays, they are buried in cemeteries in Balta Tiymez[15] and Caffa (~Feodoiya/Feodosia). The Turkish Karays go on a pilgrimage to the Chufut[16] Castle and Balta Tiymez cemetery in the Kırk-Er region in the city of Bahçesaray. They pray on their knees at the entrances and exits (1).[17] The Balta Tiymez cemetery is deeply important for the Karaites[18] since they believe that Ishak Sangari's grave is there, who was one of the missionaries of the Karâî sect who came to the Khazar Khanate. Since the Karaites never touch the trees as an act of respect, this forest was named "Balta Tiymez" (pristine) (Дубинский 2005, p. 49; Бабинов 2004, p. 15). In that forest, there are trees and oaks with roots that are believed to be the roots of their own families. In this cemetery,

every family owns a tree, and it is believed that the lineage of the family would perish if the tree dries out or perishes (Дубинский 2005, p. 50; 1) because the Karaites believe that those trees symbolize immortality and can protect them (2), which makes this place a sacred location for the Karaites; eating, drinking, speaking loudly, listening to music, singing, etc., are forbidden. The Karaites believe that national vengeance is brought upon those that are disrespectful toward the trees, and they are punished while the cult's fanatics are rewarded (Бабаджан et al. 2000, p. 26).

There are approximately 20 sacred oaks that are estimated to be 300–600 years old in the cemetery. These trees are revered by the Karaites in a certain order based on the direction of the sun. The consecration proceeds as follows: First, a Karay picks dried leaves near the oak tree that represents their family, maintains the tree, and surrounds it with a fence. Then, the Karay embraces the tree as if embracing a family member in the past after crossing the fence by their left foot. The Karay pays respect to the tree, and hugging lasts a few minutes. Thus, the Karay makes a non-verbal connection with the sacred tree (1). The cemetery is open until the sun goes down, except for Saturdays, and it is visited by the Karaites as much as possible (1). Turcologist K. Musaev provides information on this subject as follows: "Among the modern Turkish communities the Karaites are a community that conveys their ancestors' old beliefs, the beliefs such as respecting a sacred tree, faith in souls, worship to the sky-Tengri to this day" (Бабаджан et al. 2000, p. 23). According to S. Sapsal, the Hazan/religious leader states the following about the origin of this practice: "the remnants of the Karaites' belief to revere the tree which is a religious superstition is inherited from their Hazan ancestors". Tree worship, which is a continuance of Khazar traditions and a reflection of Turkish traditions, is carried out among the Chuvashs, and they are the grandchildren of Itil–Volga Bulgars (Бабаджан et al. 2000, p. 24).

After the burial, all return to the home of the deceased. Upon returning home, the relatives of the deceased light a candle. No one puts out the candle, but everyone waits for it to wane (1). When it is time to change, a new candle is lit using a candle that was lit before the previous one. It is improper for the candle to be lit on a Resting day (Sabbath/Shabbath, resting or quitting in Hebrew שַׁבָּת) (Koçak 2015, p. 430).

There is a coffin resting, probably as an influence of Islam, in the cemetery of Istanbul Karay Haskoy (Türkdoğan 2011, p. 106). The Istanbul Karaites pray for the deceased in the 1st week, 4th week, and 11th month; they commemorate the deceased and perform a ceremony resembling a mawlid with a meal. Traditionally, a horseshoe is hung at the door of the house as protection against an evil eye. A piece of iron is placed over the body of the deceased. This iron piece tradition does not exist among the Lithuanian and Crimean Karaites. Those who attend the funeral touch the coffin with a handkerchief, which they then wash along with their hands once they return home. The clock is stopped at the house of the deceased, and breaking a mirror is considered to bring bad luck. Fire, candles, or anything that brings light is lit where the deceased is (Ahmetbeyoğlu 2011). At every death anniversary, the relatives of the deceased fast in mourning for one day.[19] Fasting continues to be carried out by Lithuanian Karaites (1).

Everyone, male or female, should cover their heads at funerals, similarly to religious services. While Lithuanian Karaite men wear a red fez with tassels, women wear a head-scarf (2), which is not worn as tightly as the hijab in Islam. Among Crimean Karaites, men wear kalpaks made from merino sheep wool (Чижова 2003, pp. 81–82; Suleymanov 2012, p. 131). It is improper to speak loudly at funerals. Moreover, among Lithuanian Karaites, it is forbidden to visit other deceased relatives in the cemetery after the burial (1). The tradition of placing some pebbles equal to the number of closest relatives who could not visit the graveyard over the grave among some Karaite families is preserved. The tradition of bringing flowers to the graveyard and placing a garland over the coffin has been practiced since the end of the 19th century and the beginning of the 20th century (Чижова 2003, pp. 81–82). It is improper for pregnant women, those whose babies are under 40 days old, and those who are ill to take part in the funeral (2). If a pregnant woman enters the house of the deceased, it is believed that their child will be unhealthy. For the Karaites, it is taboo

for a pregnant woman to enter the house of the deceased person and eat or drink anything there.

Those attending the funeral touch the coffin with a handkerchief, and once they return home, they wash their hands and the handkerchief, as do the Istanbul Karaites (Чижова 2003, p. 81). Having washed their hands, they touch a tree, wall, and a stone while saying, "you will be gone by stone" and "you will be gone with a tree" (Koçak 2015, p. 430), for they want the mourning is completed. After Hazan prays at the house of the deceased, everyone is served a meal. At this gathering with a meal, women and men sit separately. The meal consists of black mourning halvah/death halvah (~kara alvas), cooked eggs, kashkaval (puff pastry with cheese), and fish (1). Black halvah is a deep expression of sharing pain and sadness. Cloves and pepper are added to the black halvah. The younger the deceased, the more pepper is added (Polkanov 1997, p. 43; 1). Eggs and fish are placed on the table; the eggs represent rebirth, and the fish represents fertility and the continuance of the generation (Çulha 2012, p. 397). After this meal, Hazan prays, and those who come to offer their condolences return home.

After the guests leave, the closest relatives and Hazan stay at the house. A black pelt or skin of an animal is laid over the floor to represent grief. Hazan sits or stands on the pelt with bread and wine in their hand. The relatives surround Hazan by kneeling or sitting. They are arranged in a line that is ordered from those that are closest to those that are furthest to the deceased with respect to their relationship. After praying for the bread and wine in hand, Hazan offers them to the blood relatives: In repeated succession, a piece of bread is offered, followed by a sip of wine, and then a piece of bread is offered again, followed by a sip of wine. When bread and wine are consumed, the ritual is considered to have ended. Mourning takes nearly a week at the house of the deceased. Father, mother, brothers, and sisters; and sons, daughters, wife, and husband mourn deeply. Every night, they go to kenesa. Women wear a black headscarf, while men wear a black belt or girdle to express their grief during the mourning period (Polkanov 1997, p. 23; 2). If the day after the funeral is Saturday, a memorial is performed for the deceased. Then, some cheese pita and some fried "cheese pita/gozleme" are prepared (1). During the first prayer at the memorial, red wine is served. The relatives of the deceased do not cook, and the guests bring food to the house of the deceased. It is improper to return the utensils that are brought by the guests right away; the cleaned utensils are returned after the seventh day of the deceased's passing (Koçak 2015, p. 430). For the meal, boiled eggs are served. They are cut in halves, and black pepper is added. Olives, black plum, and black halvah called "mourning halvah" are served, and they must be finished before dishes with meat are served. The leftovers should stay in the house rather than being thrown away. The Karay men never smoke, consume alcohol, or shave during the seven days of mourning (1; Çulha 2012, p. 387); during this period, it is unacceptable to eat meat at the house of the deceased as it is considered an act of disrespect relative to the deceased. On the seventh day, everyone visits the cemetery. The visit is adjourned until Sunday if the seventh day is Friday or Saturday. On the seventh day, the relatives first visit the grave of the deceased. When they return home, they step on black felt again and perform the "ayak-içmek" ritual again; then, they step off the felt and stand at its side. With this carried out, mourning comes to an end, thus finishing the mourning process. The Crimean Karaites call this ritual "ayaktan çıkmak" (Suleymanov 2012, p. 133). Then, they go to the kenesa and pray. After that, a meat dish is served at the house of the deceased. Mutton or lamb meat is served as the dish. The mourning halvah, kubite/kubete (a type of pastry), shepherd's roast, puff pastry with boiled meat, bean or pea sauce, wine, dried raisin, tea, or coffee are on the menu. This ceremony takes place during the week, except for Fridays and Saturdays. Taking something out of the deceased's house is forbidden for seven days. At the end of the seventh day, the relatives give fabric and money collected for the funeral to Hazan and shammashim. Hazan and shammashim must receive these gifts; otherwise, it is believed that the prayers will not be accepted, and the soul of the deceased will never find peace (2).

If Hazan and shammashim see fit, they may hand out the fabrics and money collected to the poor on the same day (2).

Both the Crimean Karaites and Lithuanian Karay Turks perform a memorial service on the 40th day (1; 2). This memorial service is performed during the daytime, which is different from the memorials that take place at night. It is performed on the earliest Saturday. Breakfast is served in the morning. Instead of black halvah or mourning halvah, "comfort halvah", which is lighter in color, is cooked. This halvah is called "Khazar halvah/Khazar katmağı". This tradition of cooking halvah among the Karaites from Crimea is still observed to this day (Дубинский 2005, p. 50). The halvah is cooked without pepper but uses plenty of honey. The leftovers are treated the same as the dishes on the seventh day (Polkanov 1994b, p. 32). Vodka is served instead of wine only during the 40th-day prayer (1). Until the 40th day, all mirrors in the house should be covered. An oil lamp is lit, and a glass of water is placed near that lamp. All covers are lifted when the tables are set for the 40th day. The glass of water is poured down into the soil or at the bottom of a tree. Pouring that glass of water onto the road is forbidden. The boiled eggs are served to the guests without being cut in half. Moreover, the guests should bring something sweet with them (1). The same rituals are carried out in the case of a stillborn or the death of a child.

The Karaites could not carry out their rituals during death and after death during the COVID-19 pandemic, which emerged in March 2020. They could only recite the forgiveness prayers from *Karaj Dińliliarniń Jalbarmach Jergialiari* (*Supplication Texts of the Karaites*) at their houses with only the relatives of the deceased after the burial (3).

A memorial service is carried out on the anniversary of the death among the Karaites. This memorial service is held for the last time on the 11th month if the deceased is male and on the 12th month if the deceased is female. Commemorating death anniversaries is a common practice among the Karaites. If the deceased is male, a last sahynč memorial is performed in the 11th month. This is called ak-kiymyak (wearing white), which is the last prayer ritual and is usually performed after 11 months on the first Saturday. What makes this ritual special is that it includes desserts such as white halvah called karaite; nuts and tahini halvah; delights; sweet tarts and cookies; and jam on the table (Polkanov 1994a, p. 25), which are served during the last prayer ritual. In addition, at the ritual of the Lithuanian Karaites, a dish with lamb or beef meat called "kıbın" and a dish made of seven layers of pastries and three layers of cheese called "katlama" are served (1), along with traditional desserts. Among the Crimean Karaites, the guests are served wine, coffee, and compote to drink (Polkanov 1994a, p. 25). However, among the Lithuanian Karaites, compote is more commonly served (1). If the deceased is a woman, the memorial takes place at the end of the 12th month. Again, the guests should bring sweet treats when they visit the memorial service (1).

Among the other tribes residing in Turkistan, there is no prohibition for dishes, including meat. Besides the meat prohibition during the mourning period among the Karaites, memorials that take place on the 3rd, 7th, and 40th day after the death of the person usually exist among other Turkish tribes, such as Tatar, Uzbek, Kazakh, Kyrgyz, Turkmen, and Chuvash (Suleymanov 2012, p. 134). Ibn Fadlan mentions the oldest mourning rituals, such as uncovering the head, crying, wailing, tearing one's clothes, etc., in the post mortem mourning traditions of pre-Islamic Turkish states. While this tradition continues with laments shouted for the merits of the deceased in Türkiye and Turkistan (Şahbaz and Taşkıran 2022, p. 48), crying, wailing, and lamenting out loud are forbidden among the Karaites. However, silently crying and shedding tears is permitted, as in Islam (2).

The tombstone is erected one year after the passing of the deceased among the Karaites. Before the tombstone is erected, the spear near the head and the stone or tree branch near the feet area, which should have been placed after the burial, are removed, and the tombstone is erected, which is different from the regular ones and bears details about the deceased. The graves of the Karaites are characteristically north–south oriented. The most common tombstones are two-horned tombstones that resemble a cradle in shape. The in-

terpretation of the tombstones' unique shape is highly intriguing. The people from whom researchers obtained firsthand experience information reported that they noticed a grave transformed into a tumulus, with pointed edges on the corners, and this is peculiar to Turkish graves; another grave with a saddle placed over the grave allegedly belonged to a warrior horseman. The tombstones usually do not have ornaments; however, occasionally, there are simple geometrical ornaments or schematic representations of trees. The inscriptions are placed at the northern point of the tombstone (Polkanov 1994a, pp. 34–38; Чижова 2003, pp. 84–85). Asceticism radically shows itself in the rejection of inscriptions on tombstones, which represents the choice of a qualified lifestyle without the temporal pleasures of this world, and it is commonly practiced among Karaites in order to reach spiritual goals. However, extensive poetic elegies are recited for the leading figures of the society. In earlier times, the inscriptions were written in Hebrew or Karaim language; on the other hand, modern transcriptions (the second half of the 20th century) are written in Russian. The inscriptions consist of the name, family bond, the date of the death, and rarely the cause of the death of the deceased, some of which have other information, such as the profession of the deceased. They also consist of an expression of blessing for the deceased; in other words, they are short farewell expressions for the deceased, which are written as an abbreviation consisting of the first letters of the words in the sentence and not as a full sentence. Writing long expressions of grief and giving details of the deceased's virtues are less common (Polkanov 1994a, pp. 34–38; Чижова 2003, pp. 84–85).

A vertical rectangular stele without a grave is erected for individuals who die in a foreign country and are buried there. This tombstone is called yolci taş, which means "wandering stone", or tikme taş, which means "erected stone". This kind of tombstone is mentioned in one of the worst curses of the Karaites: "tikme taş bol", literally meaning "may you be an erected stone", which is uttered in the sense of a curse that wishes that an individual will die in a foreign country and not in their homeland (Polkanov 1994a, pp. 34–38; Чижова 2003, pp. 84–85).

On the other hand, the modern tombstone of Crimean Karaites differs from the tombstones of other tribes, among which the Karaites are placed together only in terms of gender and certain names and surnames (Polkanov 1994a, pp. 34–38; Чижова 2003, pp. 84–85). These lines are usually engraved on the tombstone of the Karaites: "tınčlych džanyna/ džanlaryna (peace for their soul/s), tınčlych toprahyna (peace for their soil), jarych sahynč (may their memory be full of light), jachšy sahynč (may they be remembered well), jarych sahynč džanyna/ džanlaryna (may their soul/s be full of light), jachšy učmach (sacred heaven), džany/ džanlary ömiür tirliktia bolhej (may their souls be in eternal life), džany/ džanlary bah-bostanda balkyne (may their soul/s shine in heaven), sahynčy uzach tirilgiej (may their memory lasts forever), sahynčy ulusunda tirilgiej (may their memory stay alive in their community's mind), uzun sahynč syjly adyna/ adlaryna (may their name/s be remembered for a long time), etc." (Firkovičius 1999, pp. 138–39).

## 6. Discussion

The Karaites, who are considered remnants of the Khazar State (651–1048), settled in Crimea because they were exposed to the raids of Pechenegs who were fleeing from Uzes and Kipchaks. They are one of the Turkish tribes that rejected the Talmud, the orally passed down scripture of Judaism in the 8th century, and adopted Karaism. These Turks were influenced by this sect, which flourished in the Mediterranean region, and they contributed to this sect with respect to prayers, religious practices, and especially transition rituals.

Death is considered an important turning point for cultures that have a notion of an afterlife. Therefore, death is not regarded as an ending but as a transition phase to eternal life. Most traditions concerning death and the rituals performed during the funeral of Karay Turks, who are members of the Karâî sect, reflect Turkish culture and customs. Therefore, these rituals are important when it comes to passing down the traditions and customs of Karaites, functioning as a connection point between old and new generations and preserving unity and collectiveness. The Karaites have faced extinction since they are

not currently inclined toward interfaith relationships (even with other Turkish tribes) or intersecting marriages (even with Rabbanists and Krymchaks). This feature ensures that they preserve their language and identity more compared to other Turkish tribes and are more attentive toward their culture.

The food and treats that are served for the attendants of the funeral during death rituals and the use of clothing carry messages about the Karay Turkish way of life. To this day, their traditions and customs have been observed within the frames of these rituals. Similarities are observed between Crimean and Lithuanian Karaite traditions and customs regarding where the deceased is washed and shrouded and how Hazan prays. However, it is observed that post mortem rituals (on the 7th and 40th day) are preserved by the Lithuanian Karay Turks and are practiced precisely. The most important factor is that Lithuanian Karaites have formed social unity, and they are numerous in terms of population, as are their marriages. On the other hand, even though there is a decrease in practicing death rituals due to their small population and disunity, Crimean and Istanbul Karaites have not completely ceased practicing their funeral and death rituals.

**Author Contributions:** Conceptualization, E.A. and R.G.; Data curation, E.A. and R.G.; Formal analysis, E.A., R.G., E.K.I. and R.A.; Methodology, E.A., R.G. and S.E.T.; Visualization, E.K.I., R.A. and S.E.T.; Writing—original draft, E.A., E.K.I. and R.A.; Writing—review & editing, E.K.I. and R.A. All authors have read and agreed to the published version of the manuscript.

**Funding:** This research received no external funding.

**Institutional Review Board Statement:** Not applicable.

**Informed Consent Statement:** Not applicable.

**Data Availability Statement:** The data presented in this study are available within this article.

**Acknowledgments:** We thank Szymon Juchniewicz (1), Markus Lavrinovicius (2), and Diana Lavrinovicius (3) for providing information about the firsthand experience of rituals. Their information is assigned numbers, which are used as references.

**Conflicts of Interest:** The authors declare no conflict of interest.

## Notes

1  Daniel J. Lasker's *Karaism: An Introduction to the Oldest Surviving Alternative Judaism* (The Littman Library of Jewish Civilization), which gives a detailed analysis of the influences and interactions with mainstream Judaism as well as the main differences between the Karaite history, teachings and Rabbinic Judaism, is of great importance.

2  It could be written as *Tanah* since when the /k/ voiceless consonant is used in the middle of a word or at the end of a word in Hebrew, /h/ becomes fricative.

3  Tevrat is the Turkish adaptation of Arabic تورات tawrat, which means *Torah* in Hebrew.

4  Hear O Israel, the Lord our God, the Lord is one.

5  Hear O Karay, the Lord our God, the Lord is one.

6  There is a legend saying that 40 brave Karay families have come from Sarkel and Chufut Castle, and Khan wrote about them in his decrees, and they are called "40 people" (Kırklar) (Kokizov 2011, p. 873).

7  The first description of Kırk-Or or Kırk-Er belongs to Abu'l Fida, an Arab geographer who came in 1321: "Kırk-Or is in the country of Asslar (~Alanlar), which means 40 Castle in Turkish, it is a heavily fortified castle leaning on a hard-to-reach mountain. At the top of the mountain, there is an open space where the residents of the country can take shelter in case of danger" (Lebedeva 2003, p. 6).

8  A type of smooth cotton fabric.

9  In the original text, the following is stated: "Ammâ cümle Yahûdîler Karâyî mezhebinde cufud-ı cuhûdlardır. Sâ'ir Yahûdîler bu mezhebde olan cuhûdları sevmezler. Ve ta'âmlarında kaşer ve turfa nedir bilmezler. Her kimin ni'meti olursa say yağlı da olsa ve siniri çıkmamış her ne gûne et olsa yerler. Meselâ bunlar Yahûdîlerin kızılbaşlarıdır ve rûz-ı mahşer günü bunlar kızılbaşa binmezler, ammâ öbür İsrâ'ilî çufudlar yevm-i mahşerde(ki) kızılbaşa binerler derler. Mısra': 'Râfizî rûz-ı kıyâmet har buved zîr-i yehûd' demişler. Ammâ bu cufudlar gerçi İsrâ'îlî ve Mûsevîlerdir. Tevrât ve Zebûr okular, ammâ aslâ çufud lisânı bilmezler, cümle Tatarca kalpağı geyerler, şapka geymezler /…/ "(Kahraman et al. 2011, vol. VII, pp. 222–23).

10  The houses of Karay Turks usually have three windows. According to our interviewees, these windows symbolize God, the homeowner, and the guest.

[11] "I am sorry for your loss".

[12] It is an expression that literally means "may the friends get/be better", possibly in a cultural sense: "I appreciate your words, thank you". It emphasizes the wish that since the dead are gone, all the living shall live long.

[13] It is forbidden to eat meat that is cut, cooked, or had in any way during Shabbath among the Karaites. Therefore, both Lithuanian and Crimean Karaites prepare their food for Shabbath on Friday, which should be enough for consumption on Saturday and Sunday (Polkanov and Polkanova 2005, p. 89; Suleymanov 2012, p. 138; 1).

[14] The marriage contract is considered a sales contract, and the man to be married gives a certain amount of money or property to the woman's family (1).

[15] Pristine, an axe cannot touch or harm.

[16] For more details, see (Suleymanov 2012, p. 138; 1).

[17] In Crimea, Karay Turks showed their respect by dismounting while passing near the graves of the Khans in Bahcesaray for they were faithful to the tradition about the graves of the Khans in the Khazars period (Дубинский 2005, p. 50).

[18] Karaites have been in Crimea during ancient times, and this is shown by the tombstones on which Turkish names were engraved in the Karay Necropolis facing the south: The Balta-Tiymez cemetery near Chufut Castle in the years 6 and 240; in Mangup Castle in 866; in Solhat (~Old Crimea, Eski Kırım) in 910; in Kefe (~Feodosia) in 1076 and other monuments (Lebedeva 2003, p. 4). The Crimean Khanate always threatened the Karaites by destroying oak trees. When the Karaites rejected taking Timothy Khmelnitsky hostage in Chufut Castle, the Crimean Khan sent Suyun Aga as an ambassador to the Karaites. Suyun Aga threatened the Karaites angrily: "Well, know that an axe will touch Balta Tiymez!". The reason why the Karaites rejected taking Timothy Khmelnitsky hostage was that there was a blood feud about the murder of Karaimovich I, who was of Karaite descent (Бабаджан et al. 2000, pp. 26–27).

[19] The Karaites fast as a way of worship. According to the information gathered from the Karaites, there are various types of fasting: fasting during which water is allowed, fasting during which drinking water or eating any food during daytime is forbidden, votive fasting, ten-day fasting, and seven-day fasting during which it is forbidden to eat meat (2).

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
