# Peer review of "A Transition Period Ritual of the Karay Turks: Death"

_religions, doi:10.3390/rel14070870_

Round 1
Reviewer 1 Report
Unfortunately, this article is indeed a description of one sect of Karaits, but not a serious scholarship. Regarding the "Karay Turks" see Daniel J. Lasker, Karaism, (London: The Littman Library of Jewish Civilization, 2022). As Lasker describes, Seraya Shapshal (1873-1961), published a brochure in 1896 "in which he tried to demonstrate the Turko-Khazar origins of the Karaites." After fleeing from Crimea to Istanbul he was appointed as a leader of a Karaits from Lithuania where "He began his de-Judaization campaign in earnest in 1928 in order to secure a better status of his community" (Ibid pp. 80-81). Unfortunately, this article is nothing more than a summary of this sort of Karaism, without even quoting the source (or any other source). I suggest reading Lasker and the list of sources he quotes (ibid p. 84). The article needs much more work before any attempt to be published.
Author Response
We thank the reviewer for their time and effort.
We researched meticulously the death rituals among Karay Turks. One of our authors has lived in Trakai, Lithuania, where the population of the Karaites is dense, for almost a year. They gathered cultural, social, and linguistic data. Also, in preparing this manuscript, the data, and information are gathered from the works of the very Karaites and the works in the Library of Lithuania Vilnius University.
Moreover, two of the authors have done research concerning the Karaites and their papers are accepted by international academic journals before.
Karaism: An Introduction to the Oldest Surviving Alternative Judaism (The Littman Library of Jewish Civilization) by Daniel J. Lasker is published in 2022 to our knowledge. We learned about this work thanks to you, we are very grateful. Also, we have found an introductory text in Russian online http://madan.org.il/ru/news/kak-i-kogda-voznik-karaizm-kak-evreyskiy-raskol .
It will take time for us to have this work. The journal has given us 10 days to do the revisions. Therefore, we are not able to use this work. However, we have already used the information from the Karaites themselves or the works by them.
Furthermore, we aimed to summarize the Karaite sect and the origin of the Karay Turks in order not to broaden the topic. The death rituals constitute the main topic of the research.
Reviewer 2 Report
The text is exciting and necessary to shed light on a little-known tradition outside the studied context.
The detail offered enriches the reading and makes the content understandable if the subject is still being determined.
My comments seek to improve the text; therefore, I leave it up to the authors to incorporate them:
1. Cases of death at birth (stillbirth) or at an early age are not considered. Being a tradition that dates back centuries, it would be interesting to know what ritual (if different) was done in these cases or death at early ages, when they were more frequent than in today's society.
2. It is strange that the authors do not mention the Covid-19 pandemic, where funeral rites underwent radical changes quickly without respecting ancestral traditions.
3. The section dedicated to “Discussion” could be better; it does not open a discussion and results in the descriptions already written during the text.
Author Response
We thank the reviewer for their time, effort, and kind suggestions. We have taken your suggestions into account and widened our research.
We believe the title matches our manuscript's arguments. We gathered the information for Lithuanian Karaites, and we relied on the works of Yuriy Aleksandroviç Polkanov, the president of the Crimean Karaites Society, for Crimean Karaites, and concerning Istanbul Karaites we used the scarce data we had.
Szymon Juchniewicz, a Lithuanian Karaite, and Dr. Markus Lavrinovicius who is the hazan of Lithuanian Karaites have mentioned that they are in touch with both Crimean and Istanbul Karaites. Therefore, If they made a distinction between Lithuanian, Crimean, and Istanbul Karaites during gathering information from them, the rituals were described in detail according to the given distinctions. If the rituals are carried out in the same manner, they are given without pointing out the specific group.
Reviewer 3 Report
An introduction seems missing. In an introduction, an author is supposed to indicate the topic of the paper, what they will argue, and why it is relevant. This is missing: the author immediately dives into the topic. Because of this, it is unclear for me, as reader, what I’m reading and why. I advise the author to insert an introduction, clearly stating the purpose of the research. Here, also used methods need to be explained more.
Perhaps the first two paragraphs of the Discussion can be put in the Introduction: there, the author states what the purpose and background of the study has been. Then for the Discussion, I invite the author to go deeper into what is written in the last paragraph of that section, indicating how the death rituals change over time due to different factors. In my opinion, section 3 (on transition rituals) can also be placed in the Introduction, as it gives important information as to why the author focuses on death rituals.
Section 3 also needs to be rewritten, and I advise the author to delve a bit into death scholarship on transformation processes surrounding death (e.g. Hertz, Van Gennep). The statement that “human life has three important turning points titled transition periods as birth, marriage and death” is in my view incorrect: It suggests a universality that does not exist. Not all cultures recognize these same three turning points (e.g., for some, transitioning from adolescence to adulthood is more important), and turning points are not always equally important. Also the statement that death is an “inevitable ending for each living being” is incorrect and is refuted by the author themselves. Death is not an ending: it is instead a transitioning of the departed into an afterlife.
The order of section 4 seems strange: it does not follow death rituals. I advise the author to change the order of rituals, from the moment of death to moment of grieving/mourning/remembrance at cemeteries.
What I’d like the author to tease out a bit more, is the seeming paradox in the text. The author indicates that the Karaites keep close to the book as “readers of the book.” At the same time, the author shows how rituals are changing and also influenced by e.g., Turkish religious symbols. How can this seeming paradox (between being a religion of the book and being a lived religion) be explained?
I wonder if a full list of the Ten Commandments is really necessary; this is something that everybody knows or can check for themselves. Besides, if stating these 10 Commandments, I expect the author to reflect on the translation used (Bible Gateway). As stated, the Karaites are strictly committed to the original transcript at the time of the Prophet; in the text, an modern English translation is used. Can the author reflect on this?
The same goes for the long prayers. Towards the end of the chapter, the author gives long prayers. It is unclear why. What is the relevance of giving the exact words of these prayers, and not just paraphrasing them shortly?
In the abstract it is mentioned that there are “differences” because of the feature of Karaites accepting only the Torah. However, it is unclear what the differences are between: between Karaites and …??? Overall, it is also unclear who the author is really writing about: Karaites in Turkey, Crimea, Lithuania, …?? It would do well to be much more explicit in the Introduction.
In lines 281-282, the author refers to Roman Catholicism, and states that sins are not forgiven by God, but by a religious leader. This is obviously not true: the sins are forgiving by God (one asks forgiveness to God), but through the religion leader. The religious leader is the spokesperson for God, as it were, but in the end, it’s still God who forgives. This also explains the sacred nature of religious leaders in Catholicism. Is the comparison with Catholicism really relevant? If so, please correct.
At times, it is mentioned how rituals are constructed for males. But what about the females? For example, in line 371 it is stated that male graves have spears or stakes; but what about female graves? In line 484 it is said that for males, the ritual is performed in the 11th month; what about for females?
Overall, the text needs serious English editing.
Author Response
We thank the reviewer for their time and effort. Our notes are here as follows:
Necessary additions are made, and lacking information that the reviewer pointed out is added stating the methods used. Also, we edited our text language-wise.
Birth, marriage, and death transitions are important among Turkish tribes. The rituals concerning these transitions are practiced almost in the same manner and style. The difference in religious belief has no effect on practicing the rituals. For example, exactly the same rituals are carried out by both the Christian Gagauzs and the Kreshin Tatars. For Turks "transition from puberty to adolescence" is not considered as a transition period. Therefore, neither in the past nor now regarding this transition period, no ritual is practiced by any Turkish tribes.
The Karaites recite the prayers, which are found in the book titled Karaj Dińliliarniń Jalbarmach Jergialiari by M. Firkovičius (Lithuania, 1999), a Lithuanian Karaite, during birth, marriage, and death rituals.
In this study, by mentioning the difference in rituals, we mean the differences between the Karaites who live in Lithuania and the Karaites who live in Crimea. The Lithuanian Karaites outnumber the ones living in Istanbul, Kırım, and Poland. Most parts of the rituals are practiced since they form a unity socially in their geographical location. For this specific reason, the Lithuanian Karaites are deliberately chosen as our research topic. Besides, the Karaites living in Crimea, and especially in Istanbul have been influenced by Islam regardless of the scarcity of influence which we have mentioned in our paper. For example, there are gravestones in the Karaite cemetery of Hasköy in Istanbul.
Reviewer 4 Report
The discussion of the karaite tradition is interesting and will add to the larger scholarship on this relatively unknown tradition. The background and history provided introduces and draws the reader to want to know more about the Karaite. But there is something missing from this presentation. There seems to be a statement of why one would be interested in learning about how they approach death.
The description of the death rituals and mourning periods needs a little attention. Clarification is needed on which culture added to these death ritual practices. It seems that when describing a practice in one area such as Lithuania or Turkey there is mention of a different action taking place but it is not clear if this is part of the longer Karaite tradition or the adaptation of the particular culture in the place they settled. Are there specific death rituals? How does the belief in the afterlife inform these practices?
There are sweeping statements which are more conjecture than supported by evidence. For instance, line 281 about the Catholic Christian practice which is incorrect. Line 564 about the decline of Karaite.
The quality of the use of English language is good but it needs to be edited. There are some grammar errors but they are not major mistakes. The flow of the paper is okay but the transition between paragraphs needs attention.
Author Response
We thank the reviewer for their time and effort.
Necessary corrections and adjustments pointed out by the reviewer are made.
Szymon Juchniewicz, a Lithuanian Karaite, and Dr. Markus Lavrinovicius, the Hazan of the Lithuanian Karaites, have mentioned that they are in touch with the Karaites both living in Crimea and in Istanbul. Therefore, If they made any distinction concerning the rituals during gathering information, we specifically stated these; If the rituals are practiced in the same manner, we used the word "Karaites" in general.
Round 2
Reviewer 1 Report
Unfortunately, the authors did not understand my criticism. The reason why I suggested Lasker's book was not just to add him to the bibliography, but rather to give an example to a critical historical analysis. This article is very descriptive, which is good, but it is lacking any analysis, any possibility of a doubt. As I mentioned previously, the Karay Turks were a result of a very assertive creation of Seraya Shapshal. His name is still not mentioned, and nothing about his political aims. Lasker discusses it in his book, pp. 79-82, and mentions a long reference at page 82. That should not take too long to read. One does not have to accept Lasker, but his claims ought to be discussed. As I mentioned previously, a description without an analysis is not a scientific paper but just journalism or propaganda.
Author Response
We thank the reviewer for their effort and time.
We respect your comments, yet we disagree. Nonetheless, we considered your suggestions and revised our manuscript with convenient additions based on them within the spectrum of our research.
We should stress that our topic is not the ethnicity of the Karaites, but rather their rituals and religious practices concerning death. Therefore, it would not be appropriate to discuss at length the political aims or propaganda of Seraya Shapshal or any other political/historical figure.